# Investigating the p21 Ubiquitin-Independent Degron Reveals a Dual Degron Module Regulating p21 Degradation and Function

**DOI:** 10.3390/cells13191670

**Published:** 2024-10-09

**Authors:** Marianna Riutin, Pnina Erez, Julia Adler, Assaf Biran, Nadav Myers, Yosef Shaul

**Affiliations:** Department of Molecular Genetics, Weizmann Institute of Science, Rehovot P.O. Box 26, Israel; marianna.riutin@gmail.com (M.R.); pnina.dekel.erez@gmail.com (P.E.); julia.adler@weizmann.ac.il (J.A.); assafbiran@gmail.com (A.B.); nadavmyers@gmail.com (N.M.)

**Keywords:** ubiquitin-independent degradation, CDKN1A and p21WAF1/Cip1 degradation, 20S proteasome targeting

## Abstract

A group of intrinsically disordered proteins (IDPs) are subject to 20S proteasomal degradation in a ubiquitin-independent manner. Recently, we have reported that many IDPs/IDRs are targeted to the 20S proteasome via interaction with the C-terminus of the PSMA3 subunit, termed the PSMA3 Trapper. In this study, we investigated the biological significance of the IDP–Trapper interaction using the IDP p21. Using a split luciferase reporter assay and conducting detailed p21 mutagenesis, we first identified the p21 RRLIF box, localized at the C-terminus, as mediating the Trapper interaction in cells. To demonstrate the role of this box in p21 degradation, we edited the genome of HEK293 and HeLa cell lines using a CRISPR strategy. We found that the p21 half-life increased in cells with either a deleted or mutated p21 RRLIF box. The edited cell lines displayed an aberrant cell cycle pattern under normal conditions and in response to DNA damage. Remarkably, these cells highly expressed senescence hallmark genes in response to DNA damage, highlighting that the increased p21 half-life, not its actual level, regulates senescence. Our findings suggest that the p21 RRLIF box, which mediates interactions with the PSMA3 Trapper, acts as a ubiquitin-independent degron. This degron is positioned adjacent to the previously identified ubiquitin-dependent degron, forming a dual degron module that functionally regulates p21 degradation and its physiological outcomes.

## 1. Introduction

p21, also known as CDKN1A and p21^WAF1/Cip1^, regulates cell cycle checkpoints in response to many stimuli [1,2]. p21 is a largely unstructured protein [3] with diverse binding partners and, accordingly, multiple conformations. It forms complexes with and inhibits multiple cyclin-dependent kinases (CDKs), along with their associated cyclins. This broad inhibitory activity allows p21 to control multiple phases of the cell cycle [4,5]. p21 has two cyclin binding sites, one located at the N-terminus and the other at the C-terminus [6,7]. At its N-terminus, p21 forms complexes with CDK1, CDK2, CDK3, CDK4, and CDK6 along with specific cyclins associated with these kinases [4,8,9]. The p21 C-terminus contains a cyclin-binding motif [10], and also binds PCNA [11], the PSMA3 (C8) proteasome subunit [12], and REGγ [13], a cap of the 20S proteasome. The p21 C-terminus binds to the proteasome subunit C8 (PSMA3), and the C-terminal truncated p21 (p21 ΔC-terminus) was less susceptible to degradation using purified 20S proteasome [12].

p21 levels are highly regulated at both the transcriptional and posttranscriptional levels. Several studies have addressed the mechanisms of p21 proteasomal degradation. p21 can be degraded by the proteasome in a ubiquitin-dependent manner. Three E3 ubiquitin ligase complexes, SCF^SKP2^, CRL4^CDT2^, and APC/C^CDC20^ [1,14], promote the ubiquitylation and subsequent degradation of p21 when it is bound by specific protein complexes like CDK2 with PCNA or CDK1 with cyclin A or cyclin B. Interestingly, a p21 mutant lacking all lysines, the ubiquitination residue, still undergoes proteasomal degradation [15]. This process might be mediated by either ubiquitin-independent proteasomal degradation or the lysine-deficient p21 undergoing ubiquitination via its extreme N-terminus [16]. However, p21 was reported to be acetylated at its N-terminus in cells, diminishing the importance of N-terminal ubiquitination [17]. The debate over the mode of p21 degradation was also highlighted in the context of the response to UV. It has been reported that Skp2 mediates p21 ubiquitination and degradation in response to UV [18], whereas others have demonstrated that the process is Skp2- and ubiquitin-independent [19].

Different mechanisms of ubiquitin-independent proteasomal degradation of p21 have been reported. It has been demonstrated that the degradation of p21 in vivo does not necessitate the 19S proteasome cap, which is typically essential for binding to polyubiquitin chains. Instead, the REGγ proteasome activator mediates p21 degradation in vivo, indicating an alternative pathway for the p21 degradation that is independent of ubiquitination [13]. p21 is effectively degraded in vitro using the purified 20S proteasome [12,20,21,22]. Furthermore, it has been reported that the C-terminus of p21 directly binds to PSMA3 (C8), a component of alpha ring of the 20S proteasome [12,23]. Subsequent experiments by us revealed that the PSMA3 C-terminal 69 residues are sufficient to bind many intrinsically disordered proteins (IDPs) and intrinsically disordered regions (IDRs), including p21 [23]. This PSMA3 segment was accordingly termed the “substrate Trapper”.

Here, we explored the biological significance of the p21–Trapper interaction. We first identified the p21 RRLIF box at the C-terminus as crucial for binding with the Trapper. By editing the genome of HEK293 and HeLa cell lines using CRISPR, we discovered that cells with either a deleted or mutated p21 RRLIF box had an increased p21 half-life. These edited cells showed abnormal cell cycle patterns and heightened expression of senescence hallmark genes in response to DNA damage, suggesting that p21 half-life, rather than its absolute levels, may regulate senescence. The findings indicate that the p21 RRLIF box is essential for its interaction with the PSMA3 Trapper, affecting p21 stability and associated cellular outcomes.

## 2. Materials and Methods

### 2.1. Cells

HEK293 and HeLa cells were cultured in Dulbecco’s Modified Eagle Medium (DMEM) supplemented with 8% fetal bovine serum (FBS), 100 units/mL penicillin, and 100 μg/mL streptomycin. The cells were maintained in a humidified incubator at 37 °C with 5.6% CO_2_.

### 2.2. Cloning and Construct Design

Plasmids used in the study included PSMA3 and PSMA5 subunits (generously provided by Prof. K. Tanaka, Tokyo Metropolitan Institute of Medical Science, Tokyo, Japan) and 6xmyc p21 (kindly provided by Prof. Michele Pagano, NYU, New York, NY, USA). The Luc1 and Luc2 plasmids, corresponding to the N-terminal and C-terminal fragments of split Gaussia luciferase, were generously provided by Prof. A. Kimchi, Weizmann Institute of Science, Rehovot, Israel. The cloning of PSMA3 and Trapper was performed using the RF strategy [24]. The PSMA3, or the Trapper fragment (PSMA3 186–255), were cloned into the pcDNA plasmid with a 3xHA tag and Luc2. The 6xmyc p21 and Luc1 were cloned into another pcDNA plasmid. Control plasmids included tags and luciferase fragments only. Cloning of p21 mutants was achieved by PCR amplification and ligation into the backbone using restriction sites.

### 2.3. Transfection and Cell Harvest

HEK293 cells were transfected using JetPEI^®^ (Polyplus-transfection, Illkirch, France) or polyethylenimine (PEI, London, UK) 25K (Polysciences, Warrington, PA, USA) prepared at 1 mg/mL with a protocol like the commercial JetPEI reagent. HeLa cells were transfected using the jetOPTIMUS reagent (Polyplus-transfection) according to the manufacturer’s instructions. Different combinations of two plasmids were transfected per well, one encoding the receptor protein and the other encoding the p21 mutant. Alternatively, two control plasmids were transfected. In all experiments, 1 μg of each plasmid was used per well in a 6-well plate. Empty pcGN plasmid was added to bring the total DNA amount to 3 μg. Forty-eight hours after transfection, the cells were harvested and stored at −80 °C.

Cells collected from each well were lysed using a 0.1% NP-40 buffer (150 mM NaBr, 5% glycerol, 5 mM EDTA, 25 mM Tris, pH 8.5, 63.4 μM sodium oxalate, 0.6 mM reduced glutathione, 0.4 mM oxidized glutathione, 0.1% NP-40) supplemented with a protease inhibitor cocktail (Sigma-Aldrich, Saint Louis, MO, USA). Two-thirds of the lysate were used for the luciferase complementation assay, while the remaining lysate was reserved for analysis by immunoblotting. In experiments involving the proteasome inhibitor MG132, the cells were exposed to 40 μM MG132 for 4 h before harvesting. An equivalent amount of DMSO was added to the control cells at the same time.

Total RNA was extracted using TRI Reagent (MRC, Beverly Hills, CA, USA). First-strand cDNA synthesis was performed using the iScript cDNA Synthesis Kit (Quanta, Houston, TX, USA). Quantitative real-time PCR (qRT-PCR) was conducted using the LightCycler 480 (Roche, Basel, Switzerland) with PerfeCta^®^ SYBR Green FastMix (Quanta). All qPCR results were normalized to TBP1 mRNA levels.

#### 2.3.1. Immunoblot Analysis

Lysate samples were mixed with sample buffer (2% SDS, 10% glycerol, 5% 2-mercaptoethanol, 0.01% bromophenol blue, 0.0625 M Tris-HCl, pH 6.8), incubated at 95 °C for 3 min, and then loaded onto a polyacrylamide-SDS gel. After electrophoresis, the proteins were transferred to a nitrocellulose membrane, which was subsequently incubated with primary antibodies.

The primary antibodies used were mouse monoclonal anti-β-actin (Sigma-Aldrich), mouse monoclonal anti-HA (HA.11, Covance, Princeton, NJ, USA), mouse monoclonal anti-Myc (9E10; Weizmann Institute, Rehovot, Israel), and mouse monoclonal anti-p21 (F5, Santa Cruz Biotechnology, Dallas, TX, USA). Following primary antibody incubation, the membranes were incubated with an HRP-linked goat anti-mouse secondary antibody (Jackson ImmunoResearch, West Grove, PA, USA), and the corresponding signal was detected using the EZ-ECL kit (Biological Industries, Beit Haemek, Israel) on an ImageQuant™ LAS 4000 system.

Ponceau staining of the membrane was performed to confirm equal loading. Quantification of the expressed proteins and initial analysis of the membrane images were conducted using the ImageJ software.

#### 2.3.2. Luciferase Complementation Assay

Based on the protein–fragment complementation approach [25], two proteins of interest are generally fused with complementary fragments of a reporter protein. The interaction between the tested proteins brings the split fragments of the reporter protein into proximity, allowing it to regenerate into an active form. In this system, the receptor protein was fused to the C-terminal fragment of Gaussia luciferase (denoted as Luc2), while the p21 mutant or IDP peptide was fused to the N-terminal fragment of the luciferase (denoted as Luc1). Upon interaction, the luciferase enzyme was reconstituted, and its activity, which generates bioluminescence in the presence of its substrate, was detected.

The collected lysates were distributed into a 96-well white plate, with 30 μL per well. Using automated injection in a Veritas microplate luminometer (Turner BioSystems, Seattle, WA, USA), the lysates were mixed with the luciferase substrate. The substrate solution was prepared by diluting coelenterazine (Nanolight, Norman, OK, USA) to a final concentration of 20 μM in an assay buffer (25 mM Tris, pH 7.75, 1 mM EDTA, 0.6 mM reduced glutathione, 0.4 mM oxidized glutathione, and 75 mM urea). The bioluminescent signal was measured after the injection of 100 μL of the substrate solution and integrated over 10 s.

#### 2.3.3. p21-CRISPR and Construct Design

Cas9 plasmids were designed in the lab in two variations: WT-Cas9 and U3-Cas9, both based on SpCas9 from *Streptococcus pyogenes* [26]. Each plasmid contains a Flag tag, NLS, SpCas9, NLS, T2A, and mCherry regions, and is expressed under the U6 promoter. The U3 version of the Cas9 plasmid includes an additional 378 bp sequence from the UL12 subunit of HSV-1, located upstream of the SpCas9 region. A single-stranded DNA oligo (ssODN) synthesized by Merck© was used as a donor in the reaction.

Mutations were designed in the RRLIF amino acid region of the p21 protein, located on Chromosome 6, Exon 3 (36685768-36685782). A single guide RNA (TCCTCTTGGAGAAGATCAGC [CGG]) and donor oligo (TTGGCCTGGCTGACTTCTGCTGT-CTCTCCTCAGATTTCTACCACTCCAAAgctgctgctgctgctTCCAAGAGGAAGCCCTA-ATCCGCCCACAGGAAGCCTGCAGTCCTGGAAGC) were designed using the DESKGEN CRISPR tool©. To enrich and positively select for mutated cells, a co-editing strategy was employed [26,27]; briefly, this entailed editing the p21 gene while simultaneously rescuing a temperature-sensitive (ts) mutation in the essential TAF1 gene. For this purpose, temperature-sensitive HEK293 (HEK293 TAF1ts) or HeLa (HeLa TAF1ts) cells were co-transfected with both p21-editing and TAF1ts-correcting CRISPR systems and selected under restrictive temperature (39.5 °C).

#### 2.3.4. Cell Proliferation Assay (XTT Based)

p21 mutant clones were tested for their proliferation rate dependent on mitochondrial activity. A total of 2,000 cells per well were seeded in 96-well plates containing 100 µL of medium. Cells were treated with XTT reagent using a kit (Biological Industries), and the optical density (O.D.) was measured over six days. Each day, XTT reagent was diluted 1:50, and 50 µL of the mix was added to each well. After 70 min of incubation, the O.D. was measured using a Bio-Tek GEN5 microplate reader and software.

#### 2.3.5. Cell-Cycle Analysis

Propidium iodide staining for cell cycle analysis was performed on p21 mutant clones according to the lab protocol. Cells were washed with 10 mL of cold PBS and then fixed in 70% ethanol. After rehydration with PBS, RNase (50 µg/mL) and propidium iodide (25 µg/mL) were added. For FACS analysis, 30,000–50,000 cells were collected in duplicates using a BD LSRII flow cytometer (BD Biosciences, Mountain View, CA, USA) and analyzed using BD FACSDiva software (BD Biosciences).

#### 2.3.6. Trapper Binding to Peptide Array

Peptides covering the p21 sequence were prepared by Intavis Peptide (https://intavispeptides.com/en/, accessed on 20 April 2017). Various segments of PSMA and the PSMA5 C-termini were tagged with 6xHis and Flag tags. The proteins were expressed in bacteria and purified via the 6xHis tag using Ni-NTA His•Bind Resin (Novagen). The purified protein segments were then incubated with a pre-designed peptide array (Intavis, Tübingen, Germany) which was loaded with peptides derived from p21. Protein segments bound to the peptide array were detected by incubating with an anti-Flag M2 antibody (Sigma-Aldrich), followed by a horseradish peroxidase-linked goat anti-mouse antibody (Jackson ImmunoResearch). Signals were generated using the EZ-ECL kit (Biological Industries).

### 2.4. Statistical Analysis

The statistical significance (*p*-value) between means was assessed by two-tailed Student’s *t*-tests.

## 3. Results

### 3.1. p21 Interacts with the PSMA3 Trapper in the Cells

To investigate the association between p21 and the PSMA Trapper in the cells, we utilized a protein complementation approach [25]. Either full-length PSMA3 or only the Trapper region was fused to the C-terminal fragment of a luciferase, while either p21 or its fragments were fused to the N-terminal fragment of the enzyme (Figure 1B). To monitor protein levels, the constructed proteins were tagged with HA and Myc tags, respectively. Interaction between these proteins reconstructed the luciferase enzyme into its functional form, generating a detectable signal. Separated non-conjugated luciferase fragments were used as a negative control.

HEK293 cells were transiently co-transfected with different combinations of the constructed plasmids. The lysates were tested for bioluminescence (Figure 1C) and analyzed via SDS-PAGE and immunoblotting using the appropriate anti-tags antibodies (Figure 1D). The signals obtained from each PSMA3 and the Trapper together with p21 were significantly higher than the negative control, indicating that p21 interacts with both PSMA3 and the Trapper (Figure 1C). PSMA3 accumulated to a higher level than the Trapper (Figure 1D). Nevertheless, luciferase activity was lower for PSMA3-p21 pair than for the Trapper-p21 pair, implying that the Trapper alone is very active in binding p21.

The p21 C-terminal region targets the PSMA3 Trapper. To map the p21 region important for binding the Trapper, a peptide array analysis was carried out. The membrane was incubated with purified recombinant Trapper, and three positive regions were identified and labeled N, M, and C (Figure 2A). The split luciferase strategy was used to investigate which of the positive regions is functional. The negative region between M and C served as a negative control (Figure 2B). The middle region (M) was active in binding both the Trapper and the intact PSMA3, as examined by inducing luciferase activity. However, the C region was found to be more functional in terms of interaction with the Trapper and PSMA3. Next, we generated a truncated p21 mutant in which the C region was deleted and found that this mutant was inactive in inducing the luciferase activity (Figure 2C), suggesting this region is critical for binding the Trapper.

To examine whether there is an additive or synergistic effect between the three positive regions, we constructed a set of mutants and measured their activity in inducing luciferase activity, which is the readout of their capacity to interact with both the Trapper and PSMA3. We found that the M and C regions are active synergistically, especially when juxtaposed (Figure 2D). Notably, although the C fragment at the level of protein was lower than that of other constructs, it generated the highest signal in combination with the Trapper. Interestingly, in the presence of the C fragment, the Trapper showed higher accumulation (Figure 2E). Thus, the synergistic effect of the combined M and C region might be partly due to the accumulated protein level. The three p21 regions detected in panel A are highlighted in the predicted p21 structure (Figure 2F). These results suggest that the p21 C segment is sufficient to bind the PSMA3 Trapper.

### 3.2. Delineation of the Critical p21 Motif in Binding the PSMA3 Trapper

To elucidate the subregion within the p21 C fragment mediating Trapper binding, several sub-fragments were tested (Figure 3A). The constructs were assessed for their ability to bind either the Trapper (Figure 3B) or the full-length PSMA3 (Figure 3C). The analysis revealed that all the tested sub-fragments bearing the RRLIF sequence (155–159 amino acids) are active in binding the Trappe and PSMA3. Remarkably, this region overlaps with the reported cyclin binding motif [10] and with the REGg binding region [13].

The p21 N region contains a similar sequence, RRLF (Figure 3D), but lacks the isoleucine residue. Since the p21 N fragment was inactive in binding the Trapper, we hypothesized that the isoleucine residue is important for binding the Trapper and PSMA3. The p21 C fragment (131–164 aa) was mutated to I158A, altering the sequence from RRLIF to RRLAF. This mutant was then subjected to a split luciferase assay, which revealed a reduction in activity (Figure 3E), indicating a decreased affinity for binding the Trapper. The examined p21 mutants were confirmed to be properly expressed at the protein level (Figure 3F). Subsequently, the full-length p21 I158A mutant was constructed and tested, yielding similar results (Figure 3G,H). These results suggest that the RRLIF box plays a critical role in mediating 20S targeting via interaction with the PSMA3 Trapper region, with the isoleucine residue being essential for this process.

### 3.3. Analysis of Edited HEK293 Cells Highlighted the Important Role of the p21 C-Terminus in Regulating p21 Decay and Cell Growth

Having demonstrated that the p21 RRLIF box mediates the interaction between p21 and the 20S proteasome via the Trapper domain of the PSMA3 subunit, we next investigated the impact of this box on p21 decay. To this end, we utilized CRISPR technology to edit the p21 gene in HEK293 cells. Several edited clones were isolated and sequenced, with a representative clone selected for further analysis. This clone contained two differently edited alleles, resulting in two types of modified p21: a truncated version and a frameshifted version, both lacking the RRLIF box (Figure 4A). Protein expression analysis showed that both the truncated and the frameshifted p21 variants were well expressed; however, unlike the wild-type protein, the edited forms did not significantly accumulate in response to MG132 treatment, a proteasome inhibitor (Figure 4B).

To directly measure the rate of p21 decay, cells were treated with cycloheximide (CHX) to inhibit protein synthesis, allowing the observation of protein degradation over time. After 2 h, the wild-type p21 was barely detectable (Figure 4C), while levels of the mutated p21 remained unchanged (Figure 4D). These data suggest that endogenous p21 mutants lacking the RRLIF box are more stable than the wild-type protein.

p21 regulates cell cycle checkpoints by inhibiting the activity of various cyclin-dependent kinase (CDK) complexes [4,8,9]. Thus, p21 accumulation is expected to diminish cell growth, and indeed, we found that the growth of p21-edited cells was severely compromised (Figure 4E). Analysis of the cell cycle revealed that the edited cells accumulated at the G0/G1 phase (Figure 4F). Notably, cells expressing the edited p21 tend to undergo apoptosis and accumulate in the sub-G1 fraction of cell cycle profile. These results suggest that the stable p21 protein was functionally effective.

Analysis of edited HeLa cells revealed the important role of the p21 RRLIF box in regulating p21 expression. We successfully established a HeLa p21-5A mutant where the RRLIF box was substituted with AAAAA (5A) (Figure 5A). We assessed the rate of p21 decay in HeLa and HeLa p21-5A cells. The p21 protein levels were analyzed by Western blotting (Figure 5B). Since HeLa cells have low basal p21 expression, we treated them with bortezomib to inhibit proteasomal degradation 18 h before CHX treatment. Notably, under these conditions, the wild-type p21 level was much higher than that of the p21-5A mutant before CHX treatment. The lower expression of the mutant is due to reduced level of RNA. Over time, the intensity of the p21-5A band decreases more slowly compared to the wild-type p21, indicating that the mutant form is more stable and degrades at a slower rate. The half-life of the 5A mutant is significantly longer compared to that of the wild-type (Figure 5C).

Phenotypic analysis of the p21-5A cells revealed a cell growth rate comparable to that of the wild-type. Since p21 plays a role in DNA damage response, we tested the impact of p21-5A mutant under this condition. Cells were exposed to different doses of radiation (Gy) and the G0/G1 and G2/M fractions were quantified (Figure 5D). The G0/G1 fraction of HeLa p21-5A cells was significantly lower, while the G2/M fraction was higher. These findings align with the reported p21 accumulation in the G2/M phase in response to DNA damage [28,29]. Overall, these results suggest that the RRLIF box regulates p21 half-life and function.

### 3.4. Under DNA Damage Conditions, the Stable p21 Mutants Efficiently Induce the Expression of Senescence Hallmark Genes

Having demonstrated that the endogenously edited p21 protein decays more slowly than the wild-type p21 and that these stable p21 mutants impact the cell cycle, we next investigated the effect of these mutants on senescence. p21 is a key molecular mediator of DNA damage-induced senescence [1,5]. We quantified the RNA level of several genes reported to be upregulated in DNA damage-induced senescence genes in a p21-dependent manner [30]. Remarkably, the level of the tested RNAs were significantly higher in p21 gene-edited HEK293 cells (Figure 6A–F) and in HeLa p21-5A cells (Figure 6G–L). The expression patterns were very similar in both cell lines. These data suggest that the edited p21 gene, with its longer half-life, is more potent in inducing the senescence associated genes under DNA damage.

## 4. Discussion

Given the essential role of p21 in regulating cell fate and its short-lived nature, its degradation has attracted considerable attention. Several mechanisms underlying p21 proteasomal degradation have been reported, including both ubiquitin-dependent [14] and ubiquitin-independent degradation [12,17,20,23,31,32]. Ubiquitin-independent p21 degradation is also 26S proteasome-independent, occurring through the 20S-REGγ proteasomal complex [13] or the sole 20S catalytic proteasome particle [12,31]. However, details about the mechanism of p21 degradation by the 20S proteasome and its physiological relevance remain limited.

We have reported that many disordered proteins (IDPs/IDRs), including p21, directly bind the C-terminus of PSMA3, the alpha subunit of the 20S proteasome, and this interaction appears to mediate its degradation [23]. This PSMA3 segment, termed Trapper, was instrumental in identifying the interacting p21 sequence. Using split luciferase reporter assays and extensive mutagenesis studies, we identified the RRLIF p21 box as essential for binding to Trapper. Interestingly, the RRLIF box was previously reported to mediate p21 interaction with REGγ [13]. RRLIF, therefore, acts as a ubiquitin-independent degron. It is possible that this box is recognized independently by PSMA3 and REGγ, or that the RRLIF interaction with REGγ might be mediated through Trapper. Several other proteins contain the RRLIF box (Figure 7A), but whether these are active as ubiquitin-independent degrons remains to be investigated. Of particular interest is the RLIF box within ubiquitin. Whether ubiquitin is also targeted to the Trapper remains an important question awaiting investigation.

In the field of protein degradation, demonstrating the association of substrates with the proteasome in the cells is a highly challenging task. For ubiquitin-dependent degrons, although the interaction between the degron and E3 ligases in cells has been frequently observed, the direct association with the 26S proteasome is rarely shown. Our study focuses on a ubiquitin-independent degron, and we propose that it functions by directly targeting PSMA3 of the 20S proteasome. However, we have not directly demonstrated the binding of such a degron to the proteasome in cells. To the best of our knowledge, there are no previous reports showing a direct interaction between p21 and the 26S proteasome during ubiquitin-dependent degradation. Similarly, for the previously reported REGγ-dependent p21 degradation [13], no evidence has been provided to demonstrate a direct interaction between p21 and the REGγ-proteasome complex in cells.

To investigate whether the RRLIF–Trapper interaction mediates p21 degradation in the cells, we took the cell editing approach using CRISPR technology. We generated cell lines expressing either p21 with a deleted RRLIF box (HEK293 cells) or RRLIF to AAAAA mutant (p21-5A edited HeLa cells). We found that both p21 mutants exhibited longer half-lives. The simplest model suggests that the deleted mutants were poorly targeted to the Trapper and therefore escaped degradation. It is likely that these p21-edited mutants were less prone to the degradation by the 20S proteasome, although the possibility that are also less prone to degradation by the 20S-REGγ and 26S proteasome has not been ruled out.

The p21 C-terminus is involved in binding several proteins, and their mutual interplay dictates p21 function and degradation. p21 binds the proliferating cell nuclear antigen (PCNA), which is required for S phase progression [33]. This binding is mediated through a specific motif known as PCNA interaction protein (PIP) [34]. Certain residues within the p21 PIP box, along with a basic residue (R155 in p21) at the +4 position, recruits CRL4Cdt2 to facilitate ubiquitin-dependent p21 degradation, specifically during the S phase of the cell cycle or after DNA damage. This is referred to as a ubiquitin-dependent degron. Interestingly, the critical R155 residue in the ubiquitin-dependent degron is also shared by the ubiquitin-independent degron (Figure 7B) [34], together forming a double degron module.

p21 directly binds to and inhibits cyclin-dependent kinase 2 (Cdk2), playing a crucial role in regulating the cell cycle and DNA damage response. Cdk2 binds p21 at the sequence 154-KRRLIF-159 [7,10]. This p21 region overlaps with two distinct identified degrons. Notably, p21 highly accumulates when Cdk2 is overexpressed, suggesting that the Cdk2-p21 interaction may protect p21 from proteasomal degradation through both ubiquitin-dependent and -independent pathways [7,13]. In the case of ubiquitin-independent degradation, Cdk2 likely acts by sequestering p21 away from binding the Trapper or REGg, or alternatively by functioning as a Nanny protein, as previously proposed, to regulate the degradation of IDPs/IDRs [35].

The p21 protein is highly subjected to posttranslational modifications (Figure 7) that regulate not only its function but also its degradation. Notably, methylation of R156, a critical residue of the ubiquitin-independent degron, leads to p21 accumulation [36]. R156 is not part of the ubiquitin-dependent degron, suggesting that this modification selectively regulates the ubiquitin-independent p21 degradation. Another player of complexity arises from a stretch of basic amino acids in this region of p21 which regulates its nuclear import. The activity of the ubiquitin-dependent degron is confined to the nucleus [34].Whether subcellular localization also influences the activity of the ubiquitin-independent degron remains an open question.

We show that the edited cells display a cell cycle profile characteristic for a high level of p21. The edited HEK293 cells accumulated in the G0/G1 phase, and their growth rate was severely compromised. In contrast, the edited HeLa cells were less affected, with G2/M accumulation observed only in response to DNA damage. Expression of wild-type p21 (p21WT) is required for cell cycle arrest at G2 upon DNA damage [37]. The differences observed by us between the HeLa cells and the edited HEK293 cells might be due to the fact that HeLa cells contain low levels of p53 protein, the p21 regulator, due to the destabilizing effect of the HPV-18 E6 oncoprotein [38].

Functionally, p21 is known to promote the senescence of cells exposed to DNA damage [1,5,37]. Senescent cells typically express high levels of p21 [39]. In our study, even though the p21-edited gene did not exhibit higher levels of p21 compared to unedited cells under radiation, the senescence hallmark genes were still expressed. Single-cell p21-YFP analysis revealed that senescent cells maintained lower but sustained levels of p21-YFP [35]. Although the decay rate of p21-YFP was not determined, it is expected to be slower due to the chimeric nature of the protein, which combines the disordered p21 protein with the structured YFP [40]. Based on these observations, we propose that it is not the absolute level of p21, but rather its slow decay rate, which makes it sustainable and determines senescence.

## Figures and Tables

**Figure 1 cells-13-01670-f001:**
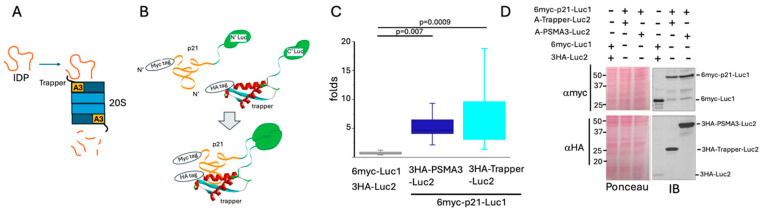
**p21 interacts with the PSMA3 Trapper.** (**A**) Schematic presentation of the proposed model whereby IDP/IDR interacts with the PSMA3 (A3) trapper as a prerequisite step for degradation by the 20S proteasome particle. (**B**) Illustration of the luciferase complementation assay system demonstrating the direct interaction between the PSMA3 Trapper and the p21 protein, both fused to luciferase split fragments. Upon interaction, a bioluminescent signal is detected. (**C**) The boxplot represents the obtained bioluminescent signals, calculated as the fold increase over the signal of the control plasmids (number of repeats N = 10). (**D**) SDS–PAGE and immunoblot analysis of the overexpressed proteins tested in panel (**C**). Ponceau staining was used as a loading control.

**Figure 2 cells-13-01670-f002:**
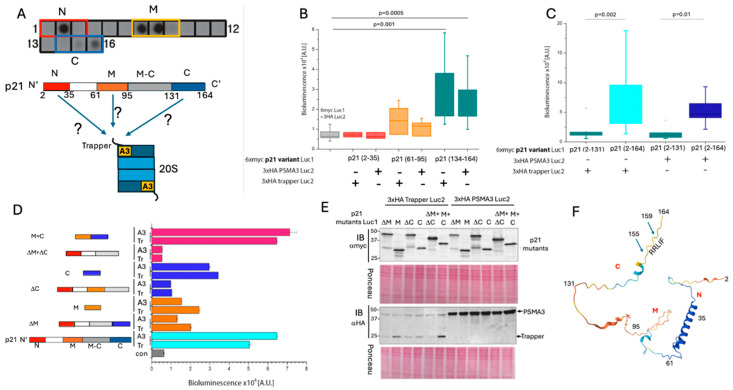
**The C-terminal segment of p21 is the most potent Trapper binding region.** (**A**) Peptide array analysis of p21 to assess which regions preferably bind the Trapper. p21 was divided into 15 amino acid–long peptides with an overlap of 5 amino acids. From left to right, each square indicates a different peptide from the N-terminus to the C-terminus of the protein. Hits that imply an interaction with the recombinant Trapper appear as darker spots. Colored frames refer to the regions that are further tested. A simplified illustration of p21 protein positive regions labeled as N, M and C. (**B**) The boxplot represents the obtained bioluminescent signals measured per pair of overexpressed positive p21 regions in binding either the Trapper or the full-length PSMA3 (N = 5). (**C**) As in panel (**B**), but truncated p21 C region deleted mutant (2–131) and full-length p21 (2–164) were tested (N = 5). (**D**) Bioluminescent signals correlating to an interaction of full-length p21, separated p21 peptides, or deletion versions of p21 with either PSMA3 (A3) or the Trapper (Tr). Error bars refer to 3 technical repetitions. Illustrated on the left are the corresponding p21 mutants (N = 3). (**E**) SDS-PAGE and immunoblot analysis of the overexpressed proteins that were tested for binding in panel (**D**). Ponceau staining was used as a loading control. (**F**) The AlphaFold predicted structure of p21, and the regions of interest are shown.

**Figure 3 cells-13-01670-f003:**
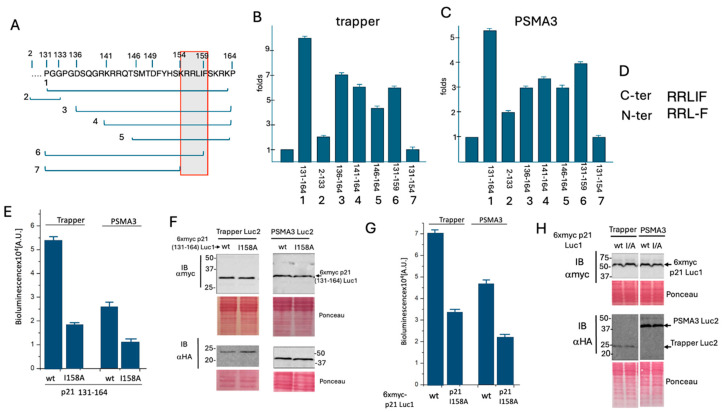
**Identification of the p21 C region sequence in mediating Trapper binding.** (**A**) Sub-fragments of the p21 C region, numbered from 1 to 7, were used for further analysis. (**B**) The fold increase in luciferase activity over the control, resulting from the transfection of the C fragment presented in panel (**A**), along with the Trapper construct, is shown (N = 3). (**C**) Similar to panel (**B**), but using the full-length PSMA3 instead of the Trapper construct (N = 3). (**D**) The RRLIF sequence within the C fragment is compared to that corresponding sequence in the N fragment. (**E**) In the context of the p21 131–164 aa C fragment, the RRLIF motif was mutated to RRLAF and subjected to the split luciferase assay with both Trapper and the PSMA3 (N = 3). (**F**) The expression levels of the plasmids used in panel (**E**) are shown, with Ponceau staining serving as a loading control. (**G**) The RRLIF box in the full-length p21 was mutated to RRLAF and analyzed with both Trapper and the PSMA3 (N = 3). (**H**) The expression level of the plasmids used in panel (**G**), with Ponceau staining as a loading control.

**Figure 4 cells-13-01670-f004:**
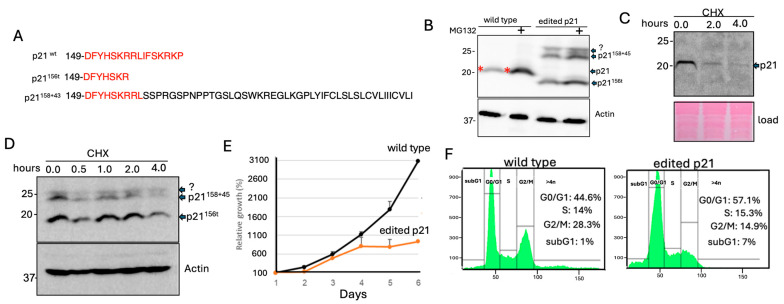
**Analysis of p21 edited HEK293 cells**. (**A**) Protein sequence of wild type p21 and the edited p21 in HEK293 cells using CRISPR technology. Two alleles were edited differently: p21^156t^ is a truncated mutant missing the last 9 amino acids of the C-terminus, including the RLIF box, while p21^158+43^ is a frameshift mutant where the C-terminus 9 amino acids are replaced by a new reading frame of 43 amino acids. The red letters in the sequence represent the wild-type p21 sequence. (**B**) p21 expression in the edited cells was analyzed in the absence or presence of MG132 for 4 h, a proteasome inhibitor. The asterisk (*) indicates the position of wild-type p21. The band labeled with a question mark (?) represents an unidentified one. (**C**) p21 protein decay was assessed using cycloheximide (CHX), a translation inhibitor, for the indicated time points. (**D**) Decay of p21 in the edited HEK293 clone, as described in panel (**A**), was examined using CHX treatment as outlined in panel. The band labeled with a question mark (?) represents an unidentified one. (**C**). (**E**) Comparison of relative growth between control HEK293 cells and the HEK293 p21-edited clone. (**F**) Cell cycle distribution of the cells was analyzed using FACS.

**Figure 5 cells-13-01670-f005:**
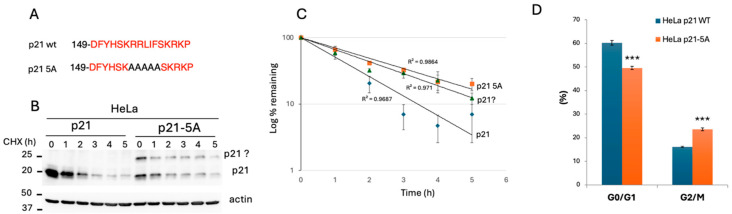
**Analysis of p21-edited HeLa cells.** (**A**) The panel displays the protein sequence of the wild-type p21 and the edited p21 in HeLa p21 cells (p21 5A), where the RRLIF box was mutated to AAAAA using CRISPR technology. (**B**) The HeLa cells and p21-5A mutant were treated with bortezomib (250 nM) for 18 h. After removing bortezomib, cells were allowed to recover for 1 h, then treated with 0.5 mM cycloheximide (CHX). Cells were harvested at specified times and analyzed by Western blotting to assess the levels of p21 and p21 5A over the CHX treatment time course. The band labeled with a question mark (p21?) represents an unidentified one. (**C**) The graph shows the degradation rate of different forms of p21 over time. The *y*-axis represents the log percentage of the remaining p21, and the *x*-axis indicates time in hours (h). Error bars reflect variability of data points at each time interval, and the R^2^ values were calculated. (**D**) Cells were irradiated (10 Gy) and the fractions of the G0/G1 and G2/M phases were quantified after 24 h. Error bars represent standard deviation or standard error of the mean. *** *p* < 0.001.

**Figure 6 cells-13-01670-f006:**
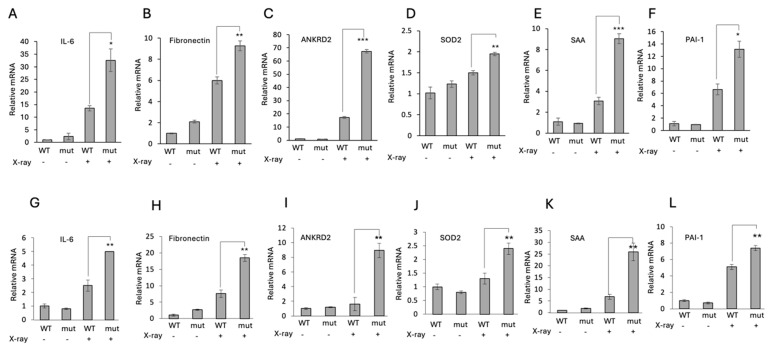
**The stable p21 mutants induce the expression of senescence-associated genes.** (**A**–**F**) the mRNA levels of several genes in wild-type and p21edited HEK293. (**G**–**L**) mRNA levels of the same set of genes in wild-type HeLa and p21–5A HeLa cells. Cells were either untreated or exposed to 10 Gy (X-ray) radiation. Error bars indicate the standard deviation or standard error of the mean. * *p* < 0.1, ** *p* < 0.01 and *** *p* < 0.001.

**Figure 7 cells-13-01670-f007:**
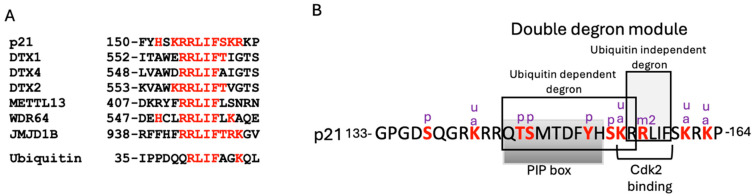
The list of proteins containing the RRLIF box and the p21 double degron composition. (**A**) Sequence alignment of proteins containing the ubiquitin-independent RRLIF degron. For each protein, the p21 sequence similarity is shown in red. (**B**) The sequence of the p21 C-terminal region, showing the functional motifs and highlighting the structure of the double degron. The potential modified residues described in PhosphoSitePlus^®^ are shown in red and marked as follows: **p** for phosphorylation sites, **u** for ubiquitination sites, **a** for acetylation sites, and **m2** for methylation sites.

## Data Availability

Data are contained within the article.

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
