# Peer review of "Investigating the p21 Ubiquitin-Independent Degron Reveals a Dual Degron Module Regulating p21 Degradation and Function"

_cells, 2024, doi:10.3390/cells13191670_

Round 1
Reviewer 1 Report
Comments and Suggestions for Authors
In this study, Ruitin et al examined the C-terminal region of p21cip1 and identified a 5-aa motif responsible for its direct proteasome interaction and ubiquitin-independent degradation. Mutation of this region in cells induced cell cycle arrest and senescence-like phenotypes.
The mechanism of p21 degradation has been a subject of long debate, to which findings of the current study may add some interesting insights. However, both the p21-proteasome interaction and the cellular phenotypes after p21 gene editing, two major components of the paper, require further experimental validation.
Major concerns:
1. Luciferase complementation was the only assay used to demonstrate p21-PSMA3 interaction, which should be backed up with other types of assays. The split luciferase system inevitably depends on exogenous expression, whereas p21 interaction with the proteasome complex (not just PSMA3) via RRLIF at the endogenous level was not demonstrated.
Along this line, S250 and S243 of the C-terminus of PSMA3 are known to be highly phosphorylated, which may be important for binding to the K/R-rich tail of p21. Are these sites also phosphorylated in isolated Trapper? Can RRLIF function as a transferable tag to recruit other proteins to the proteasome complex?
2. The CRISPR gene editing results of p21 are unimpressive. The results from 293 cells are particularly concerning due to those undesired/unknown protein products. Changing RRLIF to 5A in HeLa cells is a better choice, and yet there was still an unknown protein product. These clearly raise the question about the cleanness and reliability of the cell systems. The authors did not clarify whether the mutations were homozygous, nor did they provide detailed sequencing evidence of the edited alleles. The cell cycle responses in 293 and HeLa were distinct, which cannot be simply ascribed to the E6 oncoprotein in HeLa. At least a rescue experiment should be done to prove that the observed phenotypes were indeed caused by mutation of p21-RRLIF but not any other mystic changes of cells.
Minor points:
1. Fig. 1C and other similar data: the N of repeats should be given.
2. Line 229: “Figure 1E” should be “2E”.
3. Quality of WB can be improved.
Author Response
See the PDF doc

Reviewer 2 Report
Comments and Suggestions for Authors
In the paper titled ''p21 Ubiquitin Independent Degron Reveals a Dual Degron Module Regulating p21 Degradation and Function'' the authors perform a comprehensive study on the identification and characterization of the interaction of the tumor suppressor protein p21 with the PSMA3 subunit of the 20S proteasome, the so called trapper module. The authors demonstrate using a split luciferase model that p21 interacts with the trapper region. By performing sire directed mutagenesis the authors delineate the RRLIF sequence as the degron responsible for proteasome. These findings are further supported by the use of CRISPR gene editing to show the degron region is repsonsible for p21 stabilization and activation of cell cycle arrest function.
Overall this is a compelling and a well written and performed study that will be of general interest. It will particular interest for future studies involving proteasome degradation after 19S RP dissassociation as occurs during various stresses such as ROS and heavy metals. I recommend publication in its current format.
One small comment out of interest. The authors utilize HEK293 and HELA cells, which are well characterized. However they are transformed with adenovirus and HPV respectively, which disrupts p53 mediated tumor suppression (and p53 dependent p21 activation). Do the authors consider this to be an issue for this model?
Author Response
reviewer: One small comment out of interest. The authors utilize HEK293 and HELA cells, which are well characterized. However they are transformed with adenovirus and HPV respectively, which disrupts p53 mediated tumor suppression (and p53 dependent p21 activation). Do the authors consider this to be an issue for this model?
Response: We thank the reviewer for recognizing our manuscript's general interest. As noted, we used both cell lines to express viral genes that functionally inactivate p53 (in the HEK293 cell line) or degrade p53 (in HeLa cells). While p53 induces p21 transcription in response to DNA damage, the data on the cell cycle in HEK293 cells, figure 4F, is from cells not subjected to DNA damage, making the status of p53 irrelevant to the proposed model.
As highlighted in the text, phenotypic analysis of the HeLa p21-5A edited cells shows a growth rate comparable to wild-type cells (data not shown). Since p21 is involved in the DNA damage response, we irradiated the cells, and the results are shown in Figure 5D. In irradiated HeLa cells, p21 can still be induced by p53-independent mechanisms, including pathways involving p73 or other stress response signals.
In general, under conditions of DNA damage, p53 primarily induces p21 transcription rather than stabilizing it. Therefore, we do not believe that the observed stability of p21-5A results from the absence of functional p53.
Round 2
Reviewer 1 Report
Comments and Suggestions for Authors
The authors should include some of their responses in the final version of the paper to ease the concerns that may also arise from future readers.
The rescue experiments can be accomplished by knocking down endogenous (edited p21) while simultaneously expressing similar levels of WT or Mut p21.
Author Response
See athe ttached doc.
